# Facile electron delivery from graphene template to ultrathin metal-organic layers for boosting CO$_2$ photoreduction

Jia-Wei Wang [1,6], Li-Zhen Qiao[2,6], Hao-Dong Nie[3], Hai-Hua Huang [4], Yi Li[3], Shuang Yao[2], Meng Liu[1], Zhi-Ming Zhang [1✉], Zhen-Hui Kang [3,5] & Tong-Bu Lu[1]

Metal-organic layers with ordered structure and molecular tunability are of great potential as heterogeneous catalysts due to their readily accessible active sites. Herein, we demonstrate a facile template strategy to prepare metal-organic layers with a uniform thickness of three metal coordination layers (ca. 1.5 nm) with graphene oxide as both template and electron mediator. The resulting hybrid catalyst exhibits an excellent performance for CO$_2$ photo-reduction with a total CO yield of 3133 mmol g$^{-1}_{MOL}$ (CO selectivity of 95%), ca. 34 times higher than that of bulky Co-based metal-organic framework. Systematic studies reveal that well-exposed active sites in metal-organic layers, and facile electron transfer between heterogeneous and homogeneous components mediated by graphene oxide, greatly contribute to its high activity. This work highlights a facile way for constructing ultrathin metal-organic layers and demonstrates charge transfer pathway between conductive template and catalyst for boosting photocatalysis.

[1] Institute for New Energy Materials and Low Carbon Technologies, School of Materials Science & Engineering, Tianjin University of Technology, Tianjin 300384, China. [2] School of Chemistry and Chemical Engineering, Tianjin University of Technology, Tianjin 300384, China. [3] Institute of Functional Nano & Soft Materials (FUNSOM), Soochow University, Suzhou 215123, China. [4] KLGHEI of Environment and Energy Chemistry, School of Chemistry, Sun Yat-sen University, Guangzhou 510275, China. [5] Institute of Advanced Materials, Northeast Normal University, Changchun 103324, China. [6]These authors contributed equally: Jia-Wei Wang, Li-Zhen Qiao. ✉email: zmzhang@email.tjut.edu.cn

Sunlight-driven $CO_2$ reduction has been regarded as a promising way to simultaneously achieve solar-to-chemical energy conversion and mitigate $CO_2$ pollution[1–3]. Presently, solar-driven conversion of $CO_2$-to-chemical proceeds slowly as it requires the activation of thermodynamically stable $CO_2$ molecules and multiple electron/proton transfer processes. During the photocatalysis, various products, such as CO, $CH_4$, HCOOH, and methanol, usually generate simultaneously with low selectivity, and compete with $H_2$ evolution. To address these issues, metal-organic frameworks (MOFs), benefiting from their porous feature, ordered structure, and molecular tunability, have been widely used to mediate photocatalytic $CO_2$ reduction[4–18]. In this field, Lin et al.[16], incorporated $Re(CO)_3(bpy)Cl$ (bpy = bipyridine) modules into the UiO-67 framework, achieving a total turnover number (TON) of 10.9 in photocatalytic $CO_2$-to-CO conversion. Wang et al. achieved a high TON of 450 by using a Co-based zeolitic imidazolate framework as the cocatalyst in the presence of $[Ru(bpy)_3]^{2+}$ photosensitizer (PS)[13]. Lan et al. introduced adenine moieties into cobalt-based MOFs to drive photocatalytic $CO_2$ reduction[8]. Li et al. discovered that the amine-functionalization of linking units in a series of Fe-based MOFs can improve their photocatalytic performance for $CO_2$ reduction[15]. Recently, perovskite quantum dots were introduced into MOF matrices to synergistically catalyze $CO_2$ photoreduction[17,18]. For $CO_2$ reduction, great progress has been achieved with MOFs as low-cost catalysts. However, their performances were seriously hindered by the generally bulky nature with insufficient active sites and lethargic mass/charge transfer.

Nanosizing MOFs into ultrathin metal-organic layers (MOLs) can efficiently accelerate mass transport/electron transfer and create abundant readily accessible active sites to ensure high activity in various catalytic reactions[19–34]. Moreover, ultrathin MOFs with limited number of atoms represent ideal models to explore structure–performance relationships for further rationally constructing efficient catalysts at atomic/molecular levels. For example, a mono-carboxylic bipyridine ligand was utilized to assemble Re-/Mn-Ru molecules into monolayer Hf-based MOLs[20], the resulting Re-Ru-based MOLs can sustainably reduce $CO_2$ to CO under real sunlight for 1 week. A two-step synthesis method was used for molecular tunability to construct Ni-based MOLs, which are competent for photoreduction of diluted $CO_2$[21]. Up to date, a limited number of isolated MOLs can be synthesized due to their high surface energy and the serious lack of effective synthesis strategies. In principle, the MOLs can be stabilized by templates or surfactants to reduce the surface energy. However, these additional auxiliary components usually block the catalytic active sites to hinder efficient mass/charge transfer, thus severely reducing the catalytic activity. Therefore, rational design of ultrathin MOLs with functional substrates to integrate their advantages for synergistic photocatalysis will represent a promising method for constructing stable and highly efficient photocatalysts, but still a challenging task.

Herein, we demonstrate a facile template strategy to prepare MOLs with a uniform thickness of three metal coordination layers (ca. 1.5 nm) by using graphene oxide (GO) as both template and electron mediator. In this composite, the conductive support not only can reduce the surface energy of the ultrathin nanosheets to isolate and stabilize the three-layer MOLs, but also can efficiently accelerate electron transfer during the $CO_2$-to-CO conversion, achieving a record high CO yield of 3133 mmol $g^{-1}_{MOL}$, ca. 34 times higher than that of bulky Co-MOF, much superior to those of all the state-of-the-art MOF and MOL catalysts.

## Results

### Synthesis and characterization.
The bulky Co-MOF, $[CoL(H_2O)_2]\cdot0.5H_2O$ ($H_2L$ = 5-(1$H$-1,2,4-triazol-1-yl)isophthalic acid),

was synthesized via a solvothermal reaction of $CoCl_2\cdot6H_2O$ and $H_2L$ in DMF/$H_2O$ at 130 °C for 72 h (see the Methods for details). Single-crystal X-ray diffraction analyses reveal that Co-MOF crystallizes in a monoclinic crystal system with a space group of $C_{2/c}$ (Supplementary Table 1). As shown in Fig. 1a, the asymmetric unit of Co-MOF contains one $Co^{2+}$ cation coordinated by two aqua ligands, one N donor from the triazine moiety as well as three O atoms from two independent carboxylate groups in two L ligands. Through this coordination mode, one cobalt center connects with three organic ligands into a plane parallel to $b$ axis, forming a 2D layer-like structure. The 2D layers are stacked together via H-bonding interactions (O-H···O = 2.760(4) Å, Supplementary Fig. 1) between aqua molecules and carboxylate oxygen atoms, showing negligible voids in the framework (Supplementary Fig. 2).

Through a lot of parallel experiments, the Cd- or Zn-based MOFs with the same ligand can be prepared (see the Methods for details). It can be noticed that Cd-MOF is isostructural to Co-MOF, both exhibiting a layer-stacking structure in monoclinic crystal system with $C_{2/c}$ space group (Supplementary Fig. 3). However, Zn-MOF displays an orthorhombic crystal system with a $Pbcn$ space group. The zinc centers are in a hexa-coordinated environment completed by an aqua ligand and four L ligands into a distorted octahedral geometry, in which two equatorially coordinated organic ligands were used to link the $Zn^{2+}$ ions into an uneven 1D chain. Furthermore, these 1D chains were connected into a 2D layer by the axial coordinated L ligand. Finally, the 2D layers are fused together by the fourth L ligand into 3D structure in Zn-MOF (Supplementary Fig. 4). The above results demonstrate the facile construction of varied MOF structures based on the L ligand with different metals.

Afterwards, the bulk phase purity of Co-MOF was confirmed by powder X-ray diffraction (PXRD; Fig. 1b) by comparison with that simulated from single crystal data, and the similar conclusion can be drawn in the cases of Cd-MOF and Zn-MOF (Supplementary Figs. 5 and 6). Importantly, the sharp peak at 26.7° corresponding to (40$\bar{4}$) face was found to represent the stacking direction of MOLs with a lattice spacing of 0.33 nm, consistent with the distance between two 2D planes. Tightly stacking among these 2D layers in Co-MOF results in the bulky crystals with sizes over 50 μm, which can only be transformed into ca. 2 μm crystals after ultrasonic treatment (Supplementary Fig. 7). The close stacking will inevitably decrease the exposed active sites and impede the mass transport/electron transfer during the photocatalysis.

To overcome these problems, we attempt to use GO as 2D template to graft and stabilize metal coordination layers of Co-MOF to construct ultrathin MOL nanosheets. The synthetic procedure includes the incorporation of $Co^{2+}$ ions into GO and the subsequent in situ growth of Co-MOLs with $H_2L$ ligand on the 2D GO template (see Methods and Fig. 1c). First, both the PXRD (Fig. 1b) and FT-IR (Fig. 1d) measurements on Co-MOF, GO, and Co-MOL@GO samples reveal the effective graft of Co-MOL layers on the GO support. It should be noted that the peak of (40$\bar{4}$) face at 26.7° is absent in the PXRD pattern of Co-MOL@GO (Fig. 1b), indicating an obvious reduction of the stacking effect in MOLs on the GO support and the effective separation between different layers. Subsequently, the morphology of Co-MOL@GO was studied by transmission electron microscopy (TEM; Fig. 1e), where small nanoflakes (15–20 nm) were homogeneously distributed on the GO template. EDS mapping images indicate the even distribution of Co, N, C, and O elements on the Co-MOL@GO sample (Supplementary Fig. 8). Atomic force microscopic results also show the distribution of nanosheets on GO substrates (Fig. 1f), with an average diameter

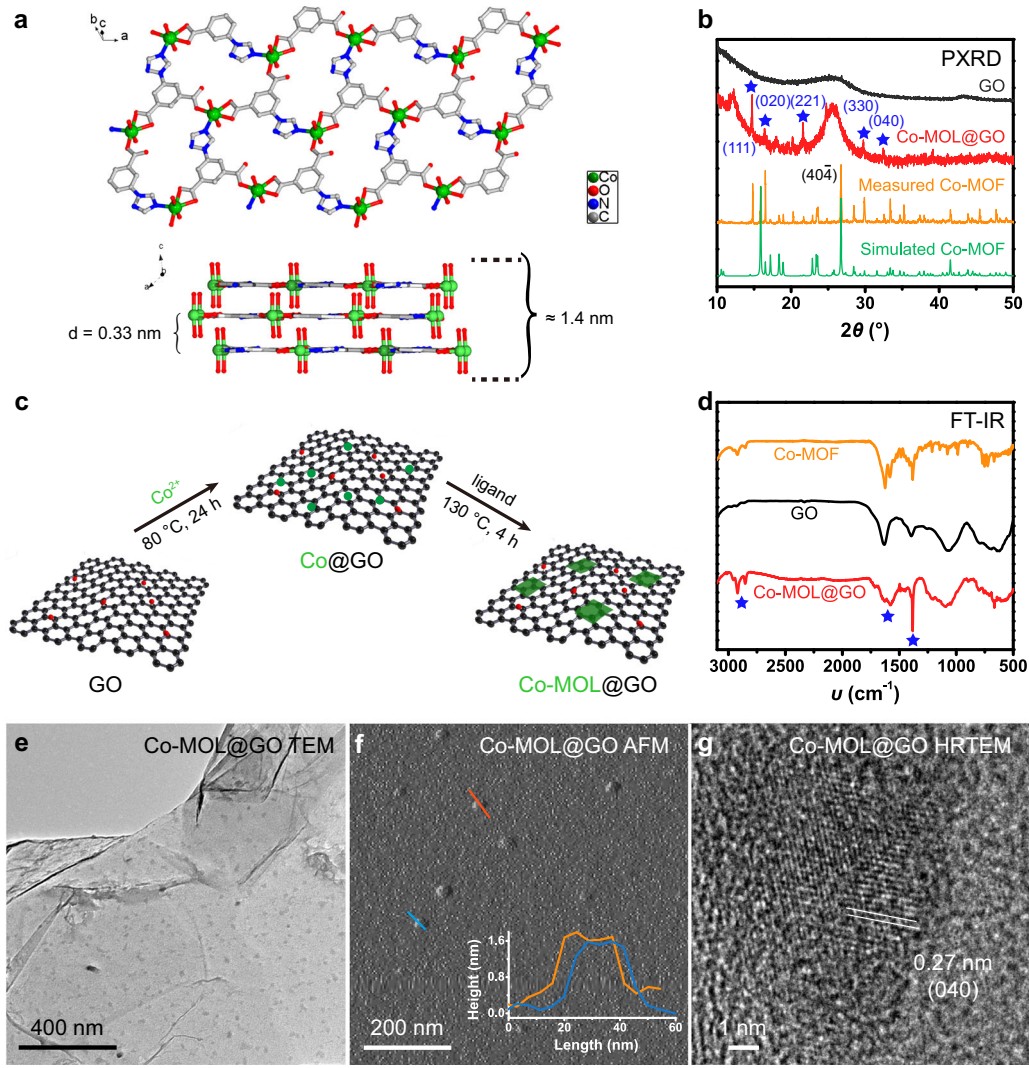

**Fig. 1 Synthesis and characterization. a** Layer-like structure of Co-MOF. **b** PXRD patterns of Co-MOF (orange), GO (black), and Co-MOL@GO (red) as well as simulated pattern of Co-MOF (green). **c** Synthesis of Co-MOL@GO. **d** FT-IR spectra of Co-MOF (orange), GO (black), and Co-MOL@GO (red). **e** TEM, **f** AFM, and **g** HRTEM results of Co-MOL@GO. The blue stars in **b** and **d** highlight the matched signals between Co-MOF and Co-MOL@GO.

of ca. 20 nm and a thickness of ca. 1.5 nm, close to three metal coordination layers of 1.4 nm determined by the single-crystal X-ray diffraction analysis (Fig. 1a). Furthermore, high-resolution TEM (Fig. 1g) measurements clearly show the crystal spacing of 0.27 nm in the tiny nanocrystal, in good agreement with the appearance of (040) facets, which is also consistent with the PXRD results with the characteristic peak located at 33.4° (0.27 nm) (Fig. 1b). Accordingly, all the above experimental results can prove the successful immobilization of ultrathin Co-MOLs on the GO template, resulting in the Co-MOL@GO composite.

The morphology and composition of MOL@GO composites can be readily tuned by varying the synthetic conditions. First, we varied the loading amount of $Co^{2+}$ to obtain corresponding Co-MOL@GO samples that were subjected to ICP-MS (Supplementary Table 2), TEM (Supplementary Fig. 9), and PXRD measurements (Supplementary Fig. 10). These results show that different sizes and amounts of Co-MOL can be grafted on the GO substrate. Then, we also tried to produce MOL@GO hybrids by loading Cd- and Zn-based MOFs on the GO substrate. Interestingly, Cd-MOL@GO can be prepared by a series of parallel experiments, as determined by corresponding PXRD and TEM results (Supplementary Figs. 5 and 11). We have adjusted a variety of experimental conditions to realize the loading of

Zn-MOF on the GO substrate; however, no nanosheets can be observed on the GO support, and the envisioned "Zn-MOF@GO" sample only showed indiscernible signals (Supplementary Fig. 6). By a detail analysis, it can be concluded that successful immobilization of ultrathin MOLs on GO depends on the crystallographic structure, in which the flat layered structures of Co/Cd-MOF should be more suitable for the co-plane π–π interaction with GO to build the 2D-2D MOL@GO composites. Overall, these results further confirm that the GO template synthesis represents a facile strategy for the synthesis of ultrathin MOL nanosheets.

With the verified morphology of Co-MOL@GO, a series of additional measurements were operated to verify the changes of Co-MOLs and GO in Co-MOL@GO sample. Initially, X-ray photoelectron spectroscopy (XPS; Fig. 2a) reveals a negative shift in the binding energies of Co 2p in Co-MOL@GO compared to those of Co-MOF. This observation indicates the existence of interactions between Co-MOF and GO surface, which can facilitate the charge transfer between Co-MOF and GO to impede the recombination of charge carriers[35]. Moreover, GO substrate was substantially reduced under the solvothermal condition, which was confirmed by the Raman spectra of Co-MOL@GO with an increased ratio between defective bands ($I_D$)

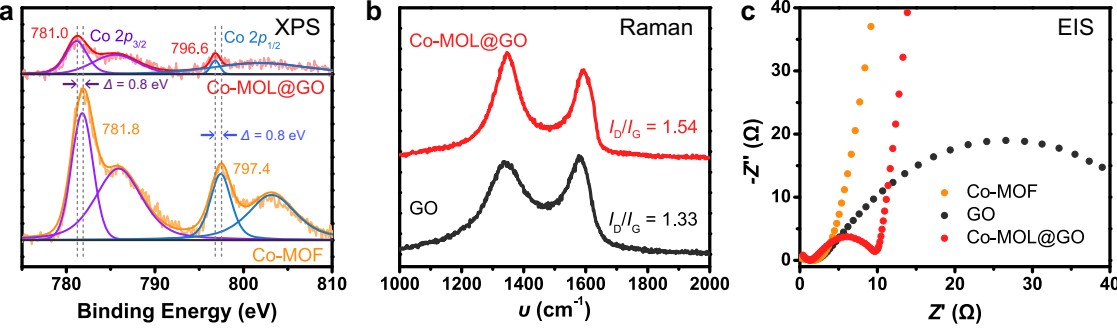

**Fig. 2 Comparison between Co-MOL@GO and Co-MOF. a** Co 2$p$ XPS spectra of Co-MOL@GO (red) and Co-MOF (orange). **b** Raman spectra of Co-MOL@GO (red) and GO (black). **c** EIS spectra of Co-MOL@GO (red), GO (black), and Co-MOF (orange).

and graphitic band ($I_G$) relative to that of GO (1.54 vs. 1.33) (Fig. 2b)[36]. The reduced GO can be more conductive and feasible for electron transfer during photocatalytic process. This conclusion was further supported by the results of electrochemical impedance spectroscopy (EIS), where Co-MOL@GO exhibits a smaller charge-transfer resistance compared to those of Co-MOF and GO (Fig. 2c). Overall, all above results suggest that Co-MOL@GO can be a good candidate as a catalyst for photocatalytic $CO_2$ reduction with its advantages in abundantly exposed active sites, excellent mass transport, and charge-transfer ability.

**Photocatalytic $CO_2$ reduction.** The catalytic performance of Co-MOL@GO for visible-light-driven ($\lambda = 450$ nm) $CO_2$ reduction was investigated in a $CO_2$-saturated $CH_3CN/H_2O$ ($v{:}v = 4{:}1$) solution. $Ru(phen)_3(PF_6)_3$ (phen = 1,10-phenanthroline; denoted as RuPS) was employed as the PS. Co-MOF, Co@GO, and GO were used as the catalysts in the control experiments. The gaseous products, CO and $H_2$, were analyzed by gas chromatography, and the liquid products, i.e., HCOOH, were checked by ion chromatography. As shown in Fig. 3 and Table 1, the main products are CO and $H_2$, and no liquid products can be detected. Remarkably, a CO yield of 216.2 mmol g$^{-1}$ with a high selectivity of 95% can be achieved during 12 h irradiation of Co-MOL@GO-based photocatalytic system, after optimization by varying the loading amount of Co-MOL (Supplementary Table 2 and Supplementary Fig. 12). In comparison, the photocatalytic experiment with bulky Co-MOF as the catalyst affords a CO yield of 91.5 mmol g$_{MOF}^{-1}$, much smaller than that of Co-MOL@GO-containing system, as well as a lower selectivity of 82%. Control experiments reveal that GO shows no activity toward $CO_2$ reduction under this photocatalytic condition, indicating that the MOLs should be the real active species in the Co-MOL@GO sample. As a result, the CO yield can be estimated as 3133 mmol g$_{MOL}^{-1}$ with the TON of 1065 ca. 34 times higher than that of bulky Co-MOF under similar condition, indicative of the remarkable intrinsic catalytic activity of ultrathin Co-MOF in Co-MOL@GO. Impressively, in terms of CO yield and selectivity, the catalytic performance of Co-MOL@GO (3133 mmol g$_{MOL}^{-1}$, 95%) is comparable to most state-of-the-art MOF catalysts for visible-light-driven $CO_2$ reduction, such as 2D-Ni$_2$TCPE[37] (20 mmol g$^{-1}$, 97%, TON 13.9), Ni MOLs[21] (25 mmol g$^{-1}$, 98%, TON 8), MAF-X27l-OH[10] (25.4 mmol g$^{-1}$, 98%, TON 2124), Co-ZIF-9[13] (209 mmol g$^{-1}$, 58%, TON 89.6), Ni(TPA/TEG)[38] (47 mmol g$^{-1}$, 99%, TON 11.5), and other examples listed in Supplementary Table 3. We further scaled up the reactor by five times to minimize the measurement error and get closer to realistic applications (see Methods and Supplementary Fig. 13 for details), which also afforded good CO yield (3467 mmol g$_{MOL}^{-1}$) and selectivity (94%) within 10 h of irradiation. In addition, the

photocatalytic system with Co@GO as the catalyst could also produce CO and $H_2$ under the same conditions, but with a much smaller amount than that of Co-MOL@GO (Supplementary Fig. 14), showing that the formation of Co-MOL nanosheets on GO is the key to achieve high-performance $CO_2$ reduction. Accordingly, GO template strategy is promising to fabricate high-performance catalysts for photocatalytic $CO_2$ reduction. These comparative results clearly demonstrate the much-enhanced intrinsic catalytic activity of 2D-nanosized Co-MOL in contrast to the bulky Co-MOF and other samples. This enhancement can be mainly attributed to its great exposure of catalytic active sites enabled by the ultrathin feature of the MOL, and the incorporation of graphene as charge-transfer mediator.

Thermal and chemical stability of Co-MOF and Co-MOL@GO were carefully investigated. For Co-MOF, thermogravimetric analysis (TGA) was conducted to investigate its thermal stability. As shown in Supplementary Fig. 15, the TGA curve shows three continuous weight losses from 95 to 320 °C, suggesting the loss of lattice and coordinated water molecules in the cavity of bulky Co-MOF. It could also be observed that a thermal decomposition of Co-MOF took place until heating up to 400 °C, revealing its high thermal stability. Meanwhile, the bulk Co-MOF was soaked in a $CO_2$-saturated $CH_3CN/H_2O$ ($v{:}v = 4{:}1$) solution containing 0.3 M TEOA, a reaction medium for photocatalytic $CO_2$ reduction. After 1 day, the solid samples were isolated for subsequent PXRD measurements. No obvious difference of the PXRD pattern can be observed compared to that of as-synthesized sample (Supplementary Fig. 16). These results demonstrate remarkable thermal and chemical stability of this Co-MOF, assuring its robustness in photocatalytic $CO_2$ reduction. For Co-MOL@GO, PXRD pattern of the solid sample isolated from the photocatalytic system shows similar signals with that of as-prepared Co-MOL@GO, indicating the intact crystalline composition of Co-MOL@GO catalyst after photocatalysis (Supplementary Fig. 17). Moreover, recycle experiments showed no substantial decrease in the activity after three runs of photocatalytic reactions, confirming the retained activity of the Co-MOL@GO catalyst (Fig. 3b). On the other hand, isotope labeling experiment with $^{13}CO_2$ shows that $^{13}CO$ is the main product in this photocatalytic system (Fig. 3c), manifesting that the CO product really derives from $CO_2$ rather than the decomposition of TEOA, RuPS, GO, or Co-MOLs. All these results demonstrate the excellent stability of Co-MOL@GO in the photocatalytic $CO_2$-to-CO conversion.

**Electron transfer pathway.** To elucidate the electron transfer pathway, the emission quenching experiments of RuPS were conducted in detail with the quenchers including Co-MOL@GO, Co-MOF, and TEOA (Fig. 3d, e and Supplementary Fig. 18). In the fluorescence spectra of RuPS, an emission peak at 598 nm was detected with the excitation at 450 nm. Upon the gradual addition

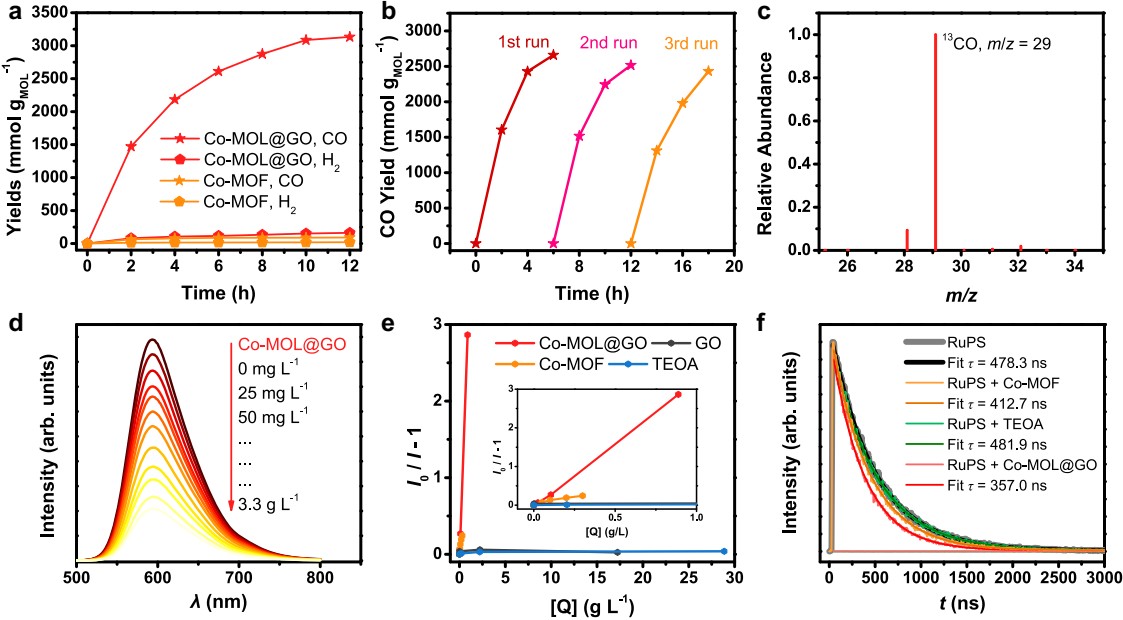

**Fig. 3 Photocatalytic CO₂ reduction. a** Time profiles of CO (star) and H₂ evolution (pentagon) catalyzed by 10 mg L⁻¹ Co-MOL@GO (red) and Co-MOF (orange) with the irradiation of a blue LED (450 nm) in CH₃CN/H₂O ($v$:$v$ = 4:1) solution under 1 atm CO₂. **b** Recycle experiments of CO production with Co-MOL@GO as the catalyst. **c** ¹³CO₂ isotope labeling experiments. **d** Fluorescence spectra of a CH₃CN/H₂O ($v$:$v$ = 4:1) solution containing 0.4 mM RuPS in the presence of 0~3.3 g L⁻¹ of Co-MOL@GO with the excitation at 450 nm. **e** Stern–Volmer plot of fluorescence by the quenchers of Co-MOL@GO (red), GO (black), Co-MOF (orange), and triethanolamine (TEOA; blue) vs. the mass concentrations ([Q]). **f** Time-resolved absorption spectra of 0.4 mM RuPS (black), RuPS with 0.3 M of TEOA (green), RuPS with 50 mg L⁻¹ of Co-MOF (orange) and RuPS with 50 mg L⁻¹ Co-MOL@GO (red) in CH₃CN with the excitation wavelength of 450 nm.

**Table 1 Photocatalytic results for CO₂ reduction to CO[a].**

| Entry | Catalyst | CO yield (mmol g⁻¹) | H₂ yield (mmol g⁻¹) | CO (%) |
|---|---|---|---|---|
| 1 | Co-MOL@GO | 216.2 (3133)[b] | 11.2 (162)[b] | 95 |
| 2 | Co-MOF | 91.5 | 19.8 | 82 |
| 3 | Co@GO | 66.7 | 3.16 | 95 |
| 4 | GO | N.D. | <0.10[c] | N.A. |
| 5 | No catalyst | N.D. | <0.10[c] | N.A. |

[a]Conditions: CO₂ saturated 5.0 mL CH₃CN/H₂O ($v$:$v$ = 4:1), catalyst (10 mg L⁻¹), RuPS (0.4 mM), TEOA (0.3 M), light intensity of 100 mW cm⁻² ($\lambda$ = 450 nm), 12 h of irradiation.
[b]Calculated based on Co-MOL moiety.
[c]Near the detection limits.

of Co-MOL@GO, the efficiency of fluorescence quenching was far higher than those obtained by addition of bulky Co-MOF and GO, indicating that the ultrathin feature of MOL and the incorporation of graphene mediator are beneficial to electron transfer. The $K_{sv}$ quenching constants of RuPS quenched by Co-MOL@GO was calculated as 3250 L g⁻¹ by Stern–Volmer plot[39,40], much higher than that obtained by addition of Co-MOF (813 L g⁻¹), and no obvious quenching can be observed in the presence of isolated GO. These results suggest the fast electron transfer between homogeneous PS* and heterogeneous Co-MOL@GO catalyst mediated by GO, affording the high catalytic performance[39,40]. Furthermore, control experiments show that negligible fluorescent quenching can be detected by addition of various amounts of TEOA (Fig. 3e), indicating that the exited RuPS* was directly quenched by Co-MOL@GO via an oxidation quenching mechanism.

The acceleration of charge transfer was confirmed by time-resolved absorption spectroscopy. As shown in Fig. 3f, the kinetic traces for excited RuPS show that the lifetime of RuPS* (478.3 ns)

was much longer than that obtained in the presence of Co-MOF (412.7 ns) catalyst, and this lifetime can be further shortened to be 357.0 ns when Co-MOL@GO composite was present. These results suggest the rapid separation and migration of photogenerated charge carriers between homogeneous PS and heterogeneous catalyst with graphene as the mediator. In the presence of TEOA, the excited lifetime of RuPS* was similar to that of isolated RuPS (481.9 vs. 478.3 ns), further confirming the oxidative quenching electron transfer pathway in this photocatalytic reaction for CO₂ reduction[41].

Furthermore, the acceleration of electron transfer mediated by GO and the roles of different components in the photocatalytic system were evaluated by in situ transient photovoltage (TPV) measurements on GO, Co-MOF, RuPS, and Co-MOL@GO. As shown in Fig. 4a, the photocurrent response of GO is the highest among the detected samples, and its curve in the CH₃CN/H₂O ($v$:$v$ = 4:1) medium is similar to that in the air (Supplementary Fig. 19). Figure 4b shows that the photocurrent intensity of Co-MOF/RuPS mixture is higher than that of isolated components of Co-MOF and RuPS. These results suggest a coupling effect between Co-MOF and RuPS that can enhance the signal, while the response of GO is still stronger than that of Co-MOF/RuPS (Fig. 4c). Then, the photocurrent intensity decreased when GO was mixed with RuPS or Co-MOF (Co-MOL), respectively. Especially, the photocurrent intensity of Co-MOL@GO composite is much lower than that from the combination of GO and RuPS (Fig. 4d). These observations infer the electron transfer pathway, in which the electrons can be transferred to both Co-MOL and RuPS from the GO surface, and Co-MOL can accept electrons more easily than RuPS.

To determine the active centers, the TPV measurements were performed in different atmospheres of N₂ and CO₂ to evaluate the real catalytic active centers (Fig. 4e–h). As shown in Fig. 4e, f, similar photocurrent intensities were observed in the curves of either GO or RuPS under N₂ or CO₂. In contrast, a sharp decrease

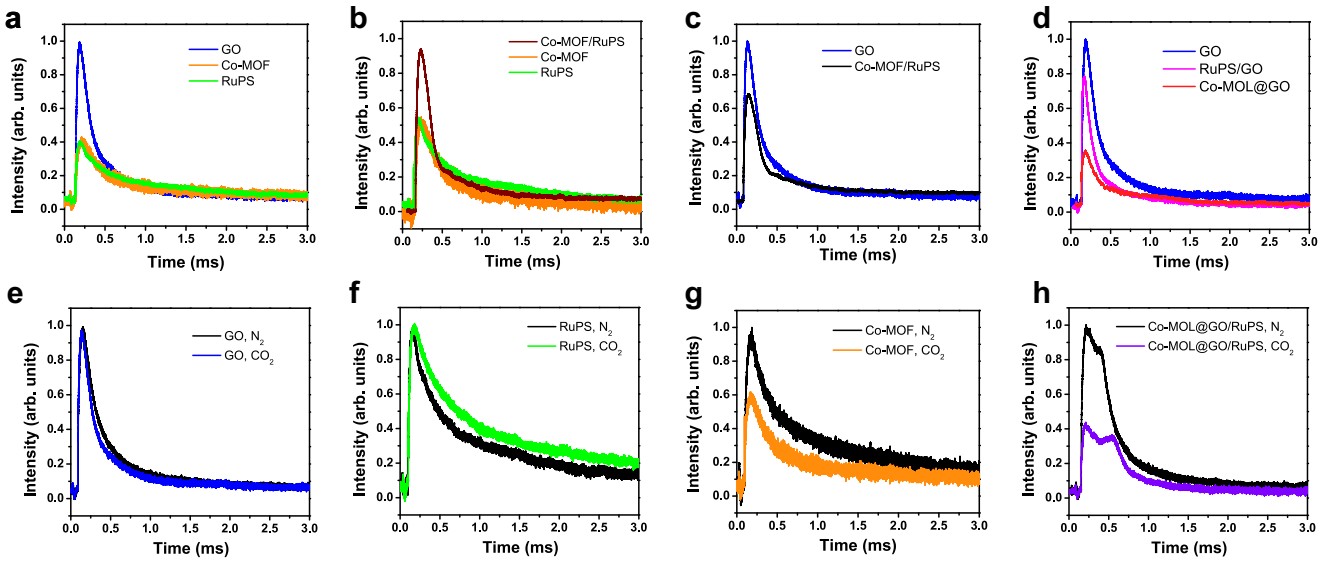

**Fig. 4 In situ transient photovoltage.** Comparison of the in situ TPV curves **a** among GO (blue), Co-MOF (orange), and RuPS (green) under N₂; **b** among Co-MOF/RuPS mixture (crimson), Co-MOF (orange), and RuPS (green) under N₂; **c** between GO (blue) and Co-MOF/RuPS mixture (black) under N₂; **d** among GO (blue), RuPS/GO mixture (pink), and Co-MOL@GO (red) under N₂. Comparison of the in situ TPV curves under N₂ (black) or CO₂ (respective color) with **e** GO (blue), **f** RuPS (green), **g** Co-MOF (orange), or **h** Co-MOL@GO/RuPS mixture (violet), respectively.

of photocurrent intensity was detected with Co-MOF under CO₂ in comparison with that under N₂ (Fig. 4g). These results reveal that the photocatalytic CO₂ reduction should occur on the surface of Co-MOF. That is, Co-MOL represents the active component in the system. To further confirm this proposal, the TPV of Co-MOL@GO/RuPS mixture was performed, and a similar trend to that of Co-MOF was observed, confirming the role of Co-MOL as the active center (Fig. 4h). Overall, the above TPV analyses confirm that Co-MOL is the active component of photocatalytic CO₂ reduction and that GO serves as the electron mediator to deliver electrons to Co-MOL.

**Mechanistic studies.** As the Co-MOL is the active component in photocatalysis, its molecular catalytic mechanism was further investigated by the electrochemical measurements. First, we studied the electrochemical behavior of GO and Co-MOL@GO loaded on the surface of the glassy carbon electrode, respectively. As shown in Supplementary Fig. 20a, the irreversible waves at ca. −0.75 V versus normal hydrogen electrode (vs. NHE) were both observed in the cyclic voltammograms (CVs) of Co-MOL@GO and GO under N₂, where the reduction currents are mainly attributed to the reduction events at GO. To avoid the interference of GO, Co-MOF was directly employed to investigate the redox behavior of the MOF catalyst. As shown in Supplementary Fig. 20b, a quasi-reversible redox couple at $E_{1/2} = -0.76$ V vs. NHE (reduction wave at −0.85 V) appeared in the CV of Co-MOF under N₂, corresponding to Co$^{II/I}$ reduction. Upon purging CO₂ into the system, an irreversible reduction wave peaking at −0.94 V vs. NHE with a relatively large current was detected, indicating a chemical step driven by Co$^{II/I}$ reduction, most possibly for catalytic CO₂ reduction[42], as the position of this reduction wave is more negative than the standard reduction potential of CO₂/CO (−0.65 V vs. NHE)[43]. Moreover, the above results indicate that the catalysis driven by the photoexcited RuPS* should be thermodynamically accessible, as the driving force from the oxidative quenching pathway ($E = -0.84$ V vs. NHE)[44] is more negative than the onset potential (ca. −0.75 V) of the catalytic wave.

According to above studies[42,45], the catalytic mechanism of Co-MOF in CO₂-to-CO conversion can be tentatively proposed, which was further verified by DFT calculation. A molecular

prototype presenting the Co-complex moiety was subjected in computational studies (Fig. 5a). As illustrated by Fig. 5, the photocatalytic cycle begins with the photo-excitation of RuPS (Fig. 5b). Then, the exited RuPS* species can be oxidatively quenched by Co-MOL@GO catalyst to drive the Co$^{II/I}$ reduction to form Co$^I$ species. The calculated potential for the reduction from Co$^{II}$ to Co$^I$ is −1.04 V vs. NHE, approaching to the measured value (ca. −0.94 V) in the presence of CO₂, further confirming the accessibility of this proposed mechanism. The Co$^I$ intermediate can react with CO₂ to generate a Co-CO₂ adduct. Then a 1e⁻/1H⁺ proton-coupled electron transfer takes place to generate a Co-COOH species (−1.09 V). Finally, the Co-COOH intermediate can release CO by cleaving the C-OH bond to recover the Co$^{II}$ state. The remaining RuPS⁺ species from the oxidative quenching pathway can be reduced to original RuPS by the TEOA, completing the photocatalytic cycle. During this photocatalytic process, the 2D GO not only serves as the template to reduce the surface energy of ultrathin nanosheets for constructing ultrathin MOLs with more exposed active sites, but also to supply conductive channels that can facilitate photoexcited electron transfer, which both play key roles in promoting the photocatalysis.

Notably, we have also operated DFT calculations on the alternate pathways for catalytic proton reduction to H₂ and CO₂ reduction to formate (Supplementary Fig. S21). It can be observed that the first protonation at the Co$^I$ intermediate is more thermodynamically unfavorable than the binding of CO₂ (28.8 vs. 10.7 kcal mol⁻¹), which should be the main reason for the selective production of CO over H₂ and formate observed in the photocatalysis with Co-MOF-based catalysts.

## Discussion

In summary, we have developed a facile and efficient strategy to construct ultrathin 2D MOLs with three metal coordination layers, where the MOLs are homogeneously distributed on the 2D GO template. The synergy between ultrathin MOL and the GO electronic conductor can greatly improve its intrinsic activity for photocatalytic CO₂ reduction. The reduced GO can act as 2D template to support and stabilize the MOLs in a uniform thickness of ca. 1.5 nm. Importantly, the GO can serve as efficient

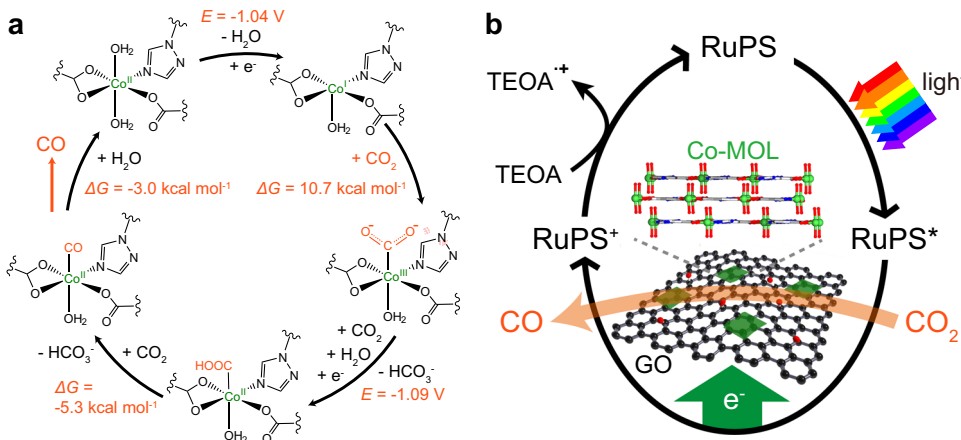

**Fig. 5 Proposed mechanisms. a** Calculated mechanism with the molecular unit of Co-MOF for photocatayltic $CO_2$-to-CO conversion, showing the calculated redox potentials and free energy changes. **b** Proposed photocatalytic mechanism with Co-MOL@GO as the catalyst.

electron mediator to bridge the gap between heterogeneous catalysts and homogeneous antenna molecules, greatly contributing to the high activity. By merging these advantages, Co-MOL@GO exhibits excellent catalytic activity and high selectivity toward visible-light-driven $CO_2$-to-CO conversion, achieving a record-high total CO yield of 3133 mmol $g_{MOL}^{-1}$ in 12 h among all the state-of-the-art MOF and MOL catalysts. This work paves an avenue on the economical preparation of ultrathin MOLs with advanced performance and demonstrates the key role of the electron mediator in dramatically promoting the photocatalysis.

## Methods
**Materials**. All the chemicals, commercially available, were used without further purification. The used water was prepared by using a Milli-Q ultrapure water purification system. Electrodes and other accessories were all purchased from Gauss Union Technology Co., Ltd.

**Instruments**. PXRD data were collected by a Smart X-ray diffractometer (SmartLab 9 KW, Rigaku, Japan) with Cu Kα radiation ($\lambda = 1.54178$ Å). EDS mapping images were acquired on an Environment Scanning Electron Microscope with a field emission gun (Quanta FEG 250, FEI, USA). The morphologies were recorded by a scanning electron microscope (Verios 460 L, FEI, USA) and a transmission electron microscope (Talos F200 X, FEI, USA). The Co/Cd contents were determined by ICP-MS (iCAP RQ, Germany). Raman spectra were recorded on a high-resolution laser confocal fiber Raman spectrometer (HORIBA EVOLVTION, HORIBA Jobin Yvon, France). More elemental information, especially for their valence states, was determined by XPS (ESCALAB250Xi, THERMO SCIENTIFIC, United Kingdom). The evolved CO and $H_2$ were monitored by using an Agilent 7820 A gas chromatography with a thermal conductivity detector (TCD) and a TDX-01 packed column, the oven temperature was held constant at 60 °C, the inlet and detector temperature were set at 80 and 200 °C, respectively. The gas analysis was also operated on another gas chromatography (GC-2014+ATF, 230 C, Shimadzu, Japan) equipped with two automated gas sampling valves, which contain a TCD and a flame ionization detector. The carrier gas used in the above two gas chromatography equipment was argon. The fluorescent spectra were conducted on a fluorescence spectrophotometer (F-7000, Hitachi, Japan). Time-resolved fluorescence measurements were measured with MicroTime 200 time-resolved confocal fluorescence instrument. The experimental data were analyzed by the SymPhoTime 64. Electrochemical measurements were operated on a CHI660D work station.

**Synthesis of Co-MOF**. In a 10-mL Teflon-lined autoclave, $CoCl_2·6H_2O$ (18.0 mg) and 5-(1$H$-1,2,4-triazol-1-yl)isophthalic acid (9.0 mg) were dissolved in a mixed solvent containing 1.75 mL DMF, 0.5 mL water, and 0.25 mL acetic acid. After a 12-h reaction under 130 °C, this mixture afforded violet crystals of Co-MOF, which were cleaned by water and ethanol, followed by drying at 60 °C.

**Synthesis of Cd-MOF**. In a 20-mL Teflon-lined autoclave, $Cd(CH_3COO)_2·2H_2O$ (30.0 mg) and 5-(1$H$-1,2,4-triazol-1-yl)isophthalic acid (24.0 mg) were dissolved in a mixed solvent containing 2.0 mL water and 0.4 mL acetic acid. After a 24-h reaction under 180 °C, this mixture afforded colorless crystals of Cd-MOF, which were cleaned by water and ethanol, followed by drying at 60 °C.

**Synthesis of Zn-MOF**. In a 20-mL Teflon-lined autoclave, $ZnCl_2$ (30.0 mg) and 5-(1$H$-1,2,4-triazol-1-yl)isophthalic acid (24.0 mg) were dissolved in a mixed solvent containing 2.0 mL water and 0.4 mL acetic acid. After a 24-h reaction under 180 °C, this mixture afforded colorless crystals of Zn-MOF, which were cleaned by water and ethanol, followed by drying at 60 °C.

**Synthesis of Co-MOL@GO**. The first step is the immobilization of $Co^{2+}$ on the 2D GO nanosheets. The GO was synthesized according to previous report[46,47]. Then, 20 mg GO and 0.5 mL water were added into 25 mL ethanol to obtain a homogeneous suspension after 1 h of ultrasonic treatment. Then, 0.1, 0.3, 0.5, or 1.0 mL aqueous solution of $CoCl_2·6H_2O$ (0.1 M) was added into this GO suspension. After reaction at 80 °C for 24 h, the resulting Co@GO powder was isolated by centrifugation, followed by washing with water for several times and drying at 60 °C. The second step is the growth of Co-MOL on the GO layers. Co@GO (20 mg) and 5-(1$H$-1,2,4-triazol-1-yl)isophthalic acid (10 mg) were added into a mixed solvent of 1.75 mL DMF, 0.5 mL water, and 0.25 mL acetic acid in a 10 mL Teflon-lined autoclave. After reaction at 130 °C for 4 h, the Co-MOL@GO sample was obtained, which was washed by DMF, water, and ethanol, followed by drying at 60 °C. The mass contents of cobalt of Co@GO and Co-MOL@GO were analyzed by ICP-MS (Supplementary Table 2). The optimized catalyst was prepared with the addition of 0.5 mL aqueous solution of $CoCl_2·6H_2O$ (0.1 M) during synthesis.

**Synthesis of Cd-MOL@GO and Zn-MOF@GO**. First, 20 mg GO and 0.5 mL water were added into 25 mL ethanol to obtain a homogeneous suspension after 1 h of ultrasonic treatment. Then, $Cd(CH_3COO)_2·2H_2O$ (30.0 mg) was added into this GO suspension. After reaction at 80 °C for 24 h, the resulting Cd@GO powder was isolated by centrifugation, followed by washing with water for two times. Without drying, the as-prepared Cd@GO and 5-(1$H$-1,2,4-triazol-1-yl)isophthalic acid (24 mg) were added into a mixed solvent of 2.0 mL water and 0.4 mL acetic acid in a 20 mL Teflon-lined autoclave. After reaction at 180 °C for 4 h, the Co-MOL@GO sample was obtained, which was washed by water and ethanol, followed by drying at 60 °C. The mass contents of cadmium of Cd@GO (18.1 ± 0.5% Cd) and Cd-MOL@GO (13.3 ± 0.2% Cd) were analyzed by ICP-MS. Accordingly, Zn-MOF@GO was intended to prepare by using $ZnCl_2$ (30.0 mg) instead of Cd salt via systematically adjusting the synthesis conditions, while the PXRD results indicate the absence of Zn-MOF crystal phase in the final sample.

**X-ray crystallography**. Single-crystal X-ray diffraction data were collected by an X-ray single crystal diffractometer (XtaLAB Pro MM003Cu/Mo, Rigaku, Japan) equipped with Cu Kα radiation ($\lambda = 1.54178$ Å). The structures were resolved using the direct method and refined on $F^2$ by the full-matrix least-squares method[48], which yields the positions of all non-hydrogen atoms and are all refined anisotropically. All hydrogen atoms of the ligand were placed in their calculated positions with fixed isotropic thermal parameters and included in the structure factor calculations in the final stage of refinement. The crystallographic data were supplied in Supplementary Table 1. The data with CCDC numbers 1965944, 2047069, and 2047070 for Co-MOF, Cd-MOF, and Zn-MOF contain the supplementary crystallographic information for this paper. The data can be obtained free of charge from The Cambridge Crystallographic Data Centre via http://www.ccdc.cam.ac.uk/.

**EIS measurements**. EIS measurements were conducted on a CHI660D electrochemical station in a conventional three-electrode cell using a Pt plate as the counter electrode and an Ag/AgCl electrode (saturated KCl) as the reference electrode. The working electrode was a catalyst-loaded fluorine-doped tin oxide (FTO) glass slide. Prior to catalyst coating, the FTO slides were cleaned by

sonication in ethanol for 30 min and dried at 353 K. The boundary of the FTO glass was protected by using Scotch tape. Then, 2 mg of catalyst was dispersed in 200 µL of ethanol with 10 µL 5% Nafion solution by sonication for 1 h to obtain a slurry, which was drop-cast onto the cleaned FTO glass. After drying overnight, the working electrode was further dried at 393 K for 5 h to improve the adhesion. EIS measurements were carried out at the open circuit potential. Prior to all measurements, the electrolyte (0.2 M $Na_2SO_4$ aqueous solution) was purged with Ar.

**Photocatalytic experiments.** The photocatalytic reduction of $CO_2$ to CO was conducted in a 17 mL home-made quartz reactor containing catalyst, $[Ru(phen)_3]$ $(PF_6)_2$, TEOA, and 5 mL $CH_3CN/H_2O$ ($v:v = 4:1$) solution under 1 atm $CO_2$ atmosphere at $293 \pm 2$ K. After purging $CO_2$ into this reaction system for 10 min, the photocatalytic reaction was initiated by irradiation of a blue LED ($\lambda = 450$ nm, light intensity = 100 mW cm$^{-2}$, irradiated area is approximate 0.5 cm$^2$). A scaling-up experiment was operated in a 67 mL home-made quartz reactor with the addition of 25 mL $CH_3CN/H_2O$ ($v:v = 4:1$, irradiation area of 2.0 cm$^2$) with identical concentrations of the above reagents. The generated gases in the headspace were analyzed by a gas chromatography, and the possible products in the solution were analyzed by an ion chromatograph. Each photocatalytic reaction was repeated three times to confirm the reliability of the data.

**In situ TPV measurements.** The working electrodes ($1 \times 2$ cm) were prepared by depositing samples (150 µL 5 mg mL$^{-1}$, dispersion liquid: 62.5% water, 25% isopropanol, and 12.5% 5 w% Nafion solution) on indium-tin oxide glass substrates. During the testing process, the working electrodes were kept dry or wet with $N_2$ or $CO_2$ saturated $CH_3CN/H_2O$ ($v:v = 4:1$), respectively. The samples were excited by a laser radiation pulse ($\lambda = 355$ nm, pulse width = 5 ns) from a third-harmonic Nd: YAG laser (Polaris II, New Wave Research, Inc.). The photocurrent is the ratio of the photovoltage to the internal resistance of the test systems.

**Cyclic voltammetry.** Cyclic voltammetric measurements were conducted on a CHI660D electrochemical station in a conventional three-electrode cell using a Pt wire as the counter electrode and another Pt wire as the pseudo reference electrode. Ferrocene (Fc) was added and its reversible couple served as the internal potential reference (vs. Fc$^+$/Fc), which was then converted to vs. NHE by adding 0.64 V[42]. Catalyst-loaded glassy carbon disk electrode with 3 mm diameter was used as the working electrode. Prior to catalyst coating, the working electrode was polished by alumina oxide powder (50 nm) for 5 min and then cleaned by sonication in ethanol for 2 min, followed by drying in air at room temperature. Then, 5 mg of catalyst was dispersed in 4.95 mL of ethanol with 50 µL 5 w% Nafion solution by sonication for 1 h to obtain a slurry. Then, 2 µL of the prepared slurry was drop-cast onto the cleaned electrode and drying under ambient conditions for 2 h. The supporting electrolyte is wet $CH_3CN$ containing tetra-$n$-butylammonium hexafluorophosphate (0.1 M).

**DFT calculation.** All the calculations were performed with Gaussian 09 program[49]. All the structures were optimized at the B3P86[50,51]/def2SVP level of theory. Based on the crystal structure, some atoms were frozen during the optimizations. Frequency analysis calculations were performed to characterize the structures to be the minima. With B3P86/def2SVP optimized geometries, the energy results were refined by single-point calculations at the B3P86/def2TZVP level of theory. The solvation effect was also evaluated with the SMD solvent model at the B3P86/def2TZVP level of theory. $H_2O$, in the presence of $CO_2$, is chosen as the real proton source in $CH_3CN/H_2O$ system, as indicated by our previous reports[45,52]. The spin variation of Co centers has been considered and the thermodynamically stable species with specific spin states were adopted as the reasonable intermediates (Supplementary Table 4). For the redox potentials calculation, the result was calculated by Eq. (1),

$$nE^\Theta F = \Delta G^\Theta_{(sol)} \tag{1}$$

where $\Delta G^\Theta_{(solv)}$ is the free energy change of the reduction process at standard conditions, $n$ is the number of electrons, $F$ is the Faraday constant. According to the experimental $Fc^{+/0}$ couple value of $-114.8$ kcal mol$^{-1}$ ($-4.98$ V)[53] in $CH_3CN$, the calculated redox potentials are referenced to $Fc^{+/0}$ by subtracting 4.98 V, then converting to vs. NHE by adding 0.64 V. It should be noted that B3P86 should be the optimal functional by the consistency between the measured $Co^{II/I}$ reduction potentials and the calculated values from a sequence of functionals in Supplementary Table 5. We also note that the small basis set, def2SVP, is sufficient and efficient to yield results approaching to the experimental values ($-1.04$ V vs. $-0.94$ V), as the full use of a larger basis set, def2TZVP, produced a similar value ($-1.02$ V) in parallel calculation while consuming a much longer calculation time.

## Data availability
Supplementary figures and data are available from the authors. The data with CCDC numbers 1965944, 2047069, and 2047070 for Co-MOF, Cd-MOF, and Zn-MOF contain the supplementary crystallographic information for this paper. The data can be obtained free of charge from The Cambridge Crystallographic Data Centre via http://www.ccdc. cam.ac.uk/. Source data are provided with this paper.

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

## Acknowledgements

This work was supported by National Natural Science Foundation of China (21722104, 21671032, 21971190, and 22071180), Natural Science Foundation of Tianjin City of China (18JCJQJC47700), China Postdoctoral Science Foundation (2020M683020), and the Fundamental Research Funds for the Central Universities (20lgpy87).

## Author contributions

J.W.W and L.Z.Q contributed equally to this work. Z.-M.Z. conceived and designed this project, J.-W.W., L.-Z.Q., H.-D.N., Y.L., and M.L. performed the experiments, H.-H.H. carried out the DFT calculation, J.-W.W., S.Y., Z.-H.K, Z.-M.Z., Z.-H.K., and T.-B.L. analyzed the data, J.-W.W., Z.-H.K., and Z.-M.Z. wrote and revised the article. All authors participated in drafting the paper, and gave approval to the final version of the manuscript.

## Competing interests

The authors declare no competing interests.
