## [Peer Review File · Nature Communications]

REVIEWER COMMENTS

Reviewer #1 (Remarks to the Author):

In this work, GO was applied as the substrate to confine the growth of Co-MOL in both directions to form a few layered Co-MOL nanoparticles. The exposed active sites of Co-MOL displayed enhanced activity and selectivity for photocatalytic CO₂ reduction to CO, compared to the bulk Co-MOL crystal. This work is recommended to be accepted after addressing a few comments raised as follows.

It is interesting to note that with the amount of Co salt doubled, much larger cubic sheets of Co-MOL were formed (Fig S5). In view of such sensitivity, has the synthesis condition been optimized to obtain best performing photocatalyst?

The interaction between Co-MOL and GO was evidenced by various materials characterization results, based on which the type of interaction (i.e., exact interface between the two materials) should be discussed. How does the interaction related to metal site and ligand site? For instance, if another metal is used, will it influence the interfacial property?

In TPV study, a 355 nm laser was used. Can you explain the reason of this light source instead of keeping the same as the light source used for reaction, LED 450 nm?

The reaction mechanism based on combined experimental data and DFT calculation appears convincing. However, the high selectivity to CO was not explained.

Finally, although based on the CO yield comparison (mmol/g), Co-MOL@GO appears to have higher activity than other photocatalysts reported in the literature, it should be noted that the absolute amount of CO produced is not directly proportional to the weight of the catalyst. In this case, only 0.05 mg of catalyst was used while most of other studies involved much larger amounts of catalyst. Such comparison is fine to be made, but the authors should not claim their catalyst is the best, unless they conduct the reaction under the same conditions. I would also like to suggest the group to scale up the reactor at least a few times in future studies, so that i) less measurement error due to larger amounts of catalyst, reagent, products, etc. ii) closer to realistic application.

Reviewer #2 (Remarks to the Author):

Given my background and expertise, I am only commenting on the computational section of the paper. I find the information given about the DFT simulations to be insufficient to understand their quality and correctness, or to reproduce the results. In particular:

- Was the spin restricted or unrestricted? Since nothing is mentioned about the treatment of the Co spin moment, it is possible that the calculations were spin-restricted, which would be incorrect given the magnetic nature of Co ions. If that's the case the computational results cannot be trusted. If the spin was unrestricted, more information is needed about the spin states considered and the procedure followed.

- Why was the functional B3P86 used?

- Was any dispersion correction applied?

- Is there any evidence that the small basis set (def2SVP) is good enough for geometry optimisation? Note that the triple zeta basis set (def2TZVP) was only used for a single-point energy correction.

Reviewer #3 (Remarks to the Author):

The article is worth of publication after the following issues have been addressed:

- The sections have to be rechecked. The conclusions are written in a section titled "Discussion" whereas the section titled "Results" is including also the discussion.
- The analytical methods and experimental details of the reactivity assessment are not enough. The GC, the detector, the column have not been specified.
- The assessment of the proposed catalytic system with the ones reported in literature has to be properly completed.

Response to Editor's and Reviewers' comments

Reviewer #1 (Remarks to the Author):

In this work, GO was applied as the substrate to confine the growth of Co-MOL in both directions to form a few layered Co-MOL nanoparticles. The exposed active sites of Co-MOL displayed enhanced activity and selectivity for photocatalytic CO₂ reduction to CO, compared to the bulk Co-MOL crystal. This work is recommended to be accepted after addressing a few comments raised as follows.

Question 1. It is interesting to note that with the amount of Co salt doubled, much larger cubic sheets of Co-MOL were formed (Fig S5). In view of such sensitivity, has the synthesis condition been optimized to obtain best performing photocatalyst?

Reply: Thank you for your kind suggestion. We have varied the Co loading of Co-MOL on GO substrate by adjusting the loading amount of Co(II) salt, which yielded four different samples subjected to the measurements of ICP-MS, PXRD, TEM and photocatalytic evaluation. The results are shown in Supplementary Table 2 and Supplementary Figure 9, 10 and 12, together with the related discussion in Page 6

and 8 in the revised manuscript. Notably, the optimal sample is the one prepared with 0.5 mL 1.0 M CoCl₂ solution based on its highest catalytic performance.

The details are as following:

The morphology and composition of MOL@GO composites can be readily tuned by varying the synthetic conditions. First, we varied the loading amount of Co²⁺ to obtain corresponding Co-MOL@GO samples which were subjected to ICP-MS (Supplementary Table 2), TEM (Supplementary Figure 9) and PXRD measurements (Supplementary Figure 10). These results show that different sizes and amounts of Co-MOL can be grafted on the GO substrate. Then, we also tried to produce MOL@GO hybrids by loading Cd- and Zn-based MOFs on the GO substrate. Interestingly, Cd-MOL@GO can be prepared by a series of parallel experiments, as determined by corresponding PXRD and TEM results (Supplementary Figure 5 and 11). We have adjusted a variety of experimental conditions to realize the loading of Zn-MOF on the GO substrate, however, no nanosheets can be observed on the GO support, and the envisioned “Zn-MOF@GO” sample only showed indiscernible signals (Supplementary Figure 6). By a detail analysis, it can be concluded that successful immobilization of ultrathin MOLs on GO depends on the crystallographic structure, in which the flat layered structures of Co/Cd-MOF should be more suitable for the co-plane π - π interaction with GO to build the 2D-2D MOL@GO composites. Overall, these results further confirm that the GO template synthesis represents a facile strategy for the synthesis of ultrathin MOL nanosheets.

Supplementary Figure 9 | TEM for Co-MOL@GO with different loading amounts. TEM images for the Co-MOL@GO samples prepared with different loading amounts of Co^{2+} , including **a** 0.1, **b** 0.3, **c** 0.5 and **d** 1.0 mL aqueous solution of $\text{CoCl}_2 \cdot 6\text{H}_2\text{O}$ (0.1 M).

Supplementary Figure 10 | PXRD results for Co-MOL@GO with different loading amounts of Co^{2+} . PXRD patterns of the Co-MOL@GO samples prepared with 0.1, 0.3, 0.5 or 1.0 mL aqueous solution of $\text{CoCl}_2 \cdot 6\text{H}_2\text{O}$ (0.1 M). The simulated PXRD pattern of Co-MOF is shown for comparison.

Supplementary Figure 12 | Photocatalysis with loading-varied Co-MOL@GO samples. Comparisons of CO and H₂ **a** yields in μmol or **b** yields in $\text{mmol g}_{\text{MOL}}^{-1}$, with 10 mg L^{-1} Co-MOL@GO samples prepared with 0.1, 0.3, 0.5 or 1.0 mL aqueous solution of $\text{CoCl}_2 \cdot 6\text{H}_2\text{O}$ (0.1 M). Other conditions: 0.4 mM RuPS and 0.3 M TEOA in 5 mL CO_2 -saturated $\text{CH}_3\text{CN}/\text{H}_2\text{O}$ (v:v = 4:1) solution. With the comprehensive estimation of CO yield, Co-MOL@GO-0.5mL was chosen as the optimal catalyst.

Supplementary Table 2 | Co amounts of Co-MOL@GO determined by ICP-MS and the photocatalytic performance.

Entry	V_{Co}	Sample	Co contents	Co-MOL	CO/H ₂	CO/H ₂ yields
	(mL) ^[a]		(w%)	contents	yields	(mol g _{MOL} ⁻¹) ^[b]
				(w%)	(μmol) ^[b]	
1	0.1	Co-MOL@ GO	0.51 ± 0.01	2.93	4.42/0.059	3017/40.3
2	0.3	Co-MOL@ GO	0.94 ± 0.02	5.40	6.67/0.082	2471/30.2
3	0.5	Co-MOL@ GO	1.20 ± 0.02	6.90	10.81/0.56	3133/162
4	1.0	Co-MOL@ GO	2.41 ± 0.06	14.8	11.11/1.19	1501/80.4

^[a] V_{Co} is the added volume of 1.0 M CoCl_2 solution.

^[b] 10 mg L^{-1} Co-MOL@GO was used for 10 h photocatalysis.

Question 2. The interaction between Co-MOL and GO was evidenced by various materials characterization results, based on which the type of interaction (i.e., exact interface between the two materials) should be discussed. How does the interaction related to metal site and ligand site? For instance, if another metal is used, will it influence the interfacial property?

Reply: Thank you for the valuable suggestion. The interaction between Co-MOL and GO is proposed to be co-plane stacking, which should be mainly attributed to the planar ligand and the flat layered structure of Co-MOF, presumably sharing a π - π interaction with GO.

To examine the above speculation, we have tried to synthesize analogous MOF and MOL@GO samples with other metal ions, including Cd^{2+} and Zn^{2+} ions. Through a lot of parallel experiments, the isolated single-crystal samples of Cd-MOF and Zn-MOF were obtained, which were analyzed by single-crystal X-ray diffraction and (Supplementary Figure 3 and 4) and powder XRD measurements (Supplementary Figure 5 and 6). It can be noticed that the Cd-MOF is isostructural to Co-MOF, both exhibiting a layer-stacking structure in a monoclinic crystal system with $C2/c$ space group (Supplementary Figure 3). However, Zn-MOF displays an orthorhombic crystal system with a $Pbcn$ space group. The zinc centers are in a hexa-coordinated environment completed by an aqua ligand and four L ligands into a distorted octahedral geometry, in which two equatorially coordinated organic ligands were used to link the Zn^{2+} ions into an uneven 1D chain. Further, these 1D chains were connected into a 2D layer by the axial coordinated L ligand. Finally, the 2D layers are fused together by the fourth L ligand into 3D structure in Zn-MOF (Supplementary Figure 4). The above results demonstrate the facile construction of varied MOF structures based on the L ligand with different metals.

Figure R1. PXRD patterns to compare the Zn-based samples.

Next, we tried to grow MOL@GO hybrids with Cd and Zn ions. Interestingly, Cd@MOL can be easily prepared, as determined by corresponding PXRD and TEM results (Supplementary Figure 5 and 11), while the sample of envision “Zn-MOL@GO” only showed indiscernible signals (Supplementary Figure 6), some of which may be attributable to the remnant ligand in the as-prepared sample (Figure R1). Corroborated with the structural features among Co-MOF, Cd-MOF and Zn-MOF, the above comparison suggests that the successful immobilization of ultrathin MOLs on GO depends on the crystallographic structure, in which the flat layered structures of Co/Cd-MOF should be more suitable for the co-plane π - π interaction with GO to build the 2D-2D MOL@GO heterojunctions.

The related discussion has been included in Page 4-6 in the main text and the synthetic methods of new samples are shown in Page 15-16 in the revised manuscript.

The details are as following:

Through a lot of parallel experiments, the Cd or Zn-based MOFs with the same ligand can be prepared (see the Methods for details). It can be noticed that Cd-MOF is

isostructural to Co-MOF, both exhibiting a layer-stacking structure in monoclinic crystal system with $C2/c$ space group (Supplementary Figure 3). However, Zn-MOF displays an orthorhombic crystal system with a $Pbcn$ space group. The zinc centers are in a hexa-coordinated environment completed by an aqua ligand and four L ligands into a distorted octahedral geometry, in which two equatorially coordinated organic ligands were used to link the Zn^{2+} ions into an uneven 1D chain. Further, these 1D chains were connected into a 2D layer by the axial coordinated L ligand. Finally, the 2D layers are fused together by the fourth L ligand into 3D structure in Zn-MOF (Supplementary Figure 4). The above results demonstrate the facile construction of varied MOF structures based on the L ligand with different metals.

Supplementary Table 1 | Crystallographic data. Crystallographic data of Co-MOF, Cd-MOF and Zn-MOF.

Complex	Co-MOF	Cd-MOF	Zn-MOF
Formula	$CoC_{10}H_9N_3O_{6.5}$	$CdC_{10}H_{10}N_3O_{6.5}$	$ZnC_{10}H_8N_3O_{5.5}$
CCDC number	1965944	2047069	2047070
Formula weight	335.14	388.61	323.56
Crystal system	monoclinic	monoclinic	orthorhombic
Space group	$C2/c$	$C2c$	$Pbcn$
Z	8	8	8
$a / \text{\AA}$	19.6624(4)	19.9580(5)	15.5913(3)
$b / \text{\AA}$	10.7223(2)	10.9485(3)	6.7209(2)
$c / \text{\AA}$	13.4748(3)	13.4505(3)	21.1187(5)
$\beta / ^\circ$	124.166(2)	122.541(2)	90.00
$V / \text{\AA}^3$	2350.55(8)	2477.66(11)	2212.98(9)
$\rho_{calcd} / \text{g m}^{-3}$	1.894	2.084	1.942
	M / mm^{-1}	14.498	3.393
2θ range collected / $^\circ$	4.94 / 79.34	9.64 / 134.14	8.38 / 158.88
Reflns collected/Indep.	8434 / 2484	7038 / 2208	7971 / 2374
R_{int}	0.0339	0.1073	0.0392
	$F(000)$	1528.0	1304.0
	GOF on f_2	1.180	1.069
Final R indices [$I >$	$^aR_1 = 0.0427, ^b wR_2 =$	$R_1 = 0.0950, wR_2 =$	$R_1 = 0.0448, wR_2 =$

$2\sigma(I)$	0.1244	0.2371	0.1340
R indices (all data)	$R_1 = 0.0449, wR_2 =$	$R_1 = 0.0970, wR_2 =$	$R_1 = 0.0477, wR_2 =$
	0.1261	0.2390	0.1377

$${}^a R_1 = \frac{\sum ||F_o| - |F_c||}{\sum F_o}, {}^b wR_2 = \left\{ \frac{\sum (F_o^2 - F_c^2)^2}{\sum w (F_o^2)^2} \right\}^{1/2}$$

Then, we also tried to produce MOL@GO hybrids by loading Cd- and Zn-based MOFs on the GO substrate. Interestingly, Cd-MOL@GO can be prepared by a series of parallel experiments, as determined by corresponding PXRD and TEM results (Supplementary Figure 5 and 11). We have adjusted a variety of experimental conditions to realize the loading of Zn-MOF on the GO substrate, however, no nanosheets can be observed on the GO support, and the envisioned “Zn-MOF@GO” sample only showed indiscernible signals (Supplementary Figure 6). By a detail analysis, it can be concluded that successful immobilization of ultrathin MOLs on GO depends on the crystallographic structure, in which the flat layered structures of Co/Cd-MOF should be more suitable for the π - π interaction with GO to build the 2D-2D MOL@GO composites. Overall, these results further confirm that the GO template synthesis represents a facile strategy for the synthesis of ultrathin MOL nanosheets.

Supplementary Figure 3 | Crystallographic structure of Cd-MOF. Presentation of crystallographic 3D structure of Cd-MOF, stacked by the 2D metal-organic layers. The H atoms are omitted for clarity.

Supplementary Figure 4 | Crystallographic structure of Zn-MOF. Presentation of crystallographic 3D structure of Zn-MOF. The H atoms are omitted for clarity.

Supplementary Figure 5 | PXRD patterns for Cd-based samples. Simulated Cd-MOF, as-prepared Cd-MOF and Cd-MOL@GO.

Supplementary Figure 11 | TEM and EDX characterizations on Cd-MOL@GO.
a TEM images of Cd-MOL@GO, showing a range of MOL diameter of 20-30 nm.
b-d EDX results of Cd-MOL@GO.

Question 3. In TPV study, a 355 nm laser was used. Can you explain the reason of this light source instead of keeping the same as the light source used for reaction, LED 450 nm?

Reply: Thank you for your question. At present, only 355, 532 and 1064 nm laser light sources are available, and the 450 nm LED cannot be simulated in our TPV measurements. On the other hand, with higher pulse frequency and intensity than the 450 nm LED light, the 355 nm laser light can also achieve the excitation of RuPS with sufficient intensity to illustrate the interface kinetics in the hybrid catalytic materials.

Question 4. The reaction mechanism based on combined experimental data and DFT calculation appears convincing. However, the high selectivity to CO was not explained.

Reply: Thank you for the valuable suggestion. We have operated DFT calculation on the alternate pathways for catalytic proton reduction to hydrogen or CO₂ reduction to formate. The corresponding energy profiles are shown in Figure 5c. It can be seen that the first protonation at the Co^I intermediate is more thermodynamically unfavorable than the binding of CO₂ (28.8 vs. 10.7 kcal mol⁻¹), which should be a main reason for the selective production of CO over H₂ and formate. The above discussion has been included in Page 13 and 14 in the revised manuscript.

The details are as following:

Notably, we have also operated DFT calculations on the alternate pathways for catalytic proton reduction to H₂ and CO₂ reduction to formate (Supplementary Figure S21). It can be observed that the first protonation at the Co^I intermediate is more thermodynamically unfavorable than the binding of CO₂ (28.8 vs. 10.7 kcal mol⁻¹), which should be a main reason for the selective production of CO over H₂ and formate observed in the photocatalysis with Co-MOF-based catalysts.

Figure 5 | Proposed mechanisms. a Calculated mechanism with the molecular unit of Co-MOF for photocatalytic CO₂-to-CO conversion, showing the calculated redox potentials and free energy changes. **b** Proposed photocatalytic mechanism with Co-MOL@GO as the catalyst.

Supplementary Figure 21 | Calculated mechanisms of Co-MOF. Calculated mechanism with the molecular unit of Co-MOF for catalytic proton reduction to H₂ and CO₂ reduction to formate, showing the calculated redox potentials and free energy changes.

Question 5. Finally, although based on the CO yield comparison (mmol g⁻¹), Co-MOL@GO appears to have higher activity than other photocatalysts reported in the literature, it should be noted that the absolute amount of CO produced is not directly proportional to the weight of the catalyst. In this case, only 0.05 mg of catalyst was used while most of other studies involved much larger amounts of catalyst. Such comparison is fine to be made, but the authors should not claim their catalyst is the best, unless they conduct the reaction under the same conditions. I would also like to suggest the group to scale up the reactor at least a few times in future studies, so that i) less measurement error due to larger amounts of catalyst, reagent, products, etc. ii) closer to realistic application.

Reply: Thank you for the kind suggestion. On one hand, we now revise our claim to avoid the statement of our catalyst is the best (Page 9). On the other hand, we scaled up the reactor to a 67 mL volume home-made quartz vessel instead of the previous 17-mL one, with the addition of five times of reagents, including 25 mL CO₂ saturated CH₃CN/H₂O (v:v = 4:1), Co-MOL@GO catalyst (10 mg L⁻¹), RuPS (0.4

mM) and TEOA (0.3 M). The time profile of H₂ and CO formation was added in the revised manuscript as Supplementary Figure 13, in which good CO yield (3467 mmol g_{MOL}⁻¹) and selectivity (94%) were still achieved within 10 h of irradiation. According to your kind suggestion, we will also use the scale up reactor in the future study. Thanks a lot!

The details are as following:

Impressively, in terms of CO yield and selectivity, the catalytic performance of Co-MOL@GO (3133 mmol g_{MOL}⁻¹, 95%) is comparable to most state-of-the-art MOF catalysts for visible-light-driven CO₂ reduction, such as 2D-Ni₂TCPE³⁷ (20 mmol g⁻¹, 97%, TON 13.9), Ni MOLs²¹ (25 mmol g⁻¹, 98%, TON 8), MAF-X271-OH¹⁰ (25.4 mmol g⁻¹, 98%, TON 2124), Co-ZIF-9¹³ (209 mmol g⁻¹, 58%, TON 89.6), Ni(TPA/TEG)³⁸ (47 mmol g⁻¹, 99%, TON 11.5) and other examples listed in Supplementary Table 3). We further scaled up the reactor by five times to minimize the measurement error and get closer to realistic applications (see Methods and Supplementary Figure 13 for details), which also afforded good CO yield (3467 mmol g_{MOL}⁻¹) and selectivity (94%) within 10 h of irradiation.

Supplementary Figure 13 | Photocatalysis. Time profiles of CO (black star) and H₂ (red pentagon) evolution catalyzed by 10 mg L⁻¹ Co-MOL@GO in a five-times scaling-up reaction system.

Reviewer #2 (Remarks to the Author):

Given my background and expertise, I am only commenting on the computational section of the paper. I find the information given about the DFT simulations to be insufficient to understand their quality and correctness, or to reproduce the results. In particular:

Question 1. Was the spin restricted or unrestricted? Since nothing is mentioned about the treatment of the Co spin moment, it is possible that the calculations were spin-restricted, which would be incorrect given the magnetic nature of Co ions. If that's the case the computational results cannot be trusted. If the spin was unrestricted, more information is needed about the spin states considered and the procedure followed.

Reply: Thank you for your kind suggestion. We here present the Supplementary Table 4 to display the calculation results on the intermediates at different spin states, in which the thermodynamically stable species with specific spin states are adopted as the reasonable intermediates in Figure 5. The discussion has been included in Methods (Page 19).

The details are as following:

Supplementary Table 4 | Relative free energy in kcal mol⁻¹ for intermediates at different spin states.^a

Entry	Intermediates	singlet	doublet	triplet	quartet
1	Co ^{II}	N.A.	23.32	N.A.	0
2	Co ^I	22.62	N.A.	0	N.A.
3	Co ^{II} -CO ₂	N.A.	0.08	N.A.	0
4	Co ^{II} -COOH	N.A.	0	N.A.	0.05
5	Co ^{II} -CO	N.A.	11.28	N.A.	0
6	Co ^{III} -H	7.85	N.A.	0	N.A.
7	Co ^{II} -H	N.A.	1.13	N.A.	0
8	Co ^{II} -H ₂	N.A.	20.97	N.A.	0

9	Co ^{II} -HCOO	N.A.	18.30	N.A.	0
---	------------------------	------	-------	------	---

[a] For each lowest energy spin state, the free energy is set as reference point, 0 kcal mol⁻¹.

Figure 5 | Proposed mechanisms. **a** Calculated mechanism with the molecular unit of Co-MOF for photocatalytic CO₂-to-CO conversion, showing the calculated redox potentials and free energy changes. **b** Proposed photocatalytic mechanism with Co-MOL@GO as the catalyst.

DFT calculation

All the calculations were performed with Gaussian 09 program.⁴⁹ All the structures were optimized at the B3P86^{50,51}/def2SVP level of theory. Based on the crystal structure, some atoms were frozen during the optimizations. Frequency analysis calculations were performed to characterize the structures to be the minima. With B3P86/def2SVP optimized geometries, the energy results were refined by single-point calculations at the B3P86/def2TZVP level of theory. The solvation effect was also evaluated with the SMD solvent model at the B3P86/def2TZVP level of theory. H₂O,

in the presence of CO₂, is chosen as the real proton source in CH₃CN/H₂O system, as

indicated by our previous reports.^{45,52} The spin variation of Co centers has been

considered and the thermodynamically stable species with specific spin states were

adopted as the reasonable intermediates (Supplementary Table 4). For the redox

potentials calculation, the result was calculated by,

$$nE^{\ominus}F = \Delta G_{sol}$$

()

where $\Delta G^\ominus(\text{solv})$ is the free energy change of the reduction process at standard conditions, n is the number of electrons, F is the Faraday constant. According to the experimental $\text{Fc}^{+/0}$ couple value of $-114.8 \text{ kcal mol}^{-1}$ (-4.98 V)⁵³ in CH_3CN , the calculated redox potentials are referenced to $\text{Fc}^{+/0}$ by subtracting 4.98 V, then

converting to versus NHE by adding 0.64 V. It should be noted that B3P86 should be the optimal functional by the consistency between the measured $\text{Co}^{\text{II/I}}$ reduction potentials and the calculated values from a sequence of functionals in Supplementary Table 5. We also note that the small basis set, def2SVP, is sufficient and efficient to yield results approaching to the experimental values (-1.04 V vs. -0.94 V), as the full use of a larger basis set, def2TZVP, produced a similar value (-1.02 V) in parallel calculation while consuming a much longer calculation time.

Question 2. Why was the functional B3P86 used?

Reply: We previously found that B3P86 should be the optimal functional according to its minimum consistency between the measured $\text{Co}^{\text{II/I}}$ reduction potentials and the calculated values from a sequence of functionals in Supplementary Table 5. The discussion has been included in Methods (Page 19).

The details are as following:

Supplementary Table 5 | Calculated $\text{Co}^{\text{II/I}}$ reduction potentials by different functional with def2SVP basis set.

Entry	Method/Functional	$\text{Co}^{\text{II/I}}$ reduction potential (V vs. NHE)
1	Measured value	-0.94
2	B3P86	-1.04
3	B3P86-D3	-1.16
4	M06-D3-	-1.60
5	M06-L-D3	-1.42
7	M06	-1.57
8	M06-L	-1.41

9	B3LYP	-1.56
10	B3LYP-D3	-1.68

Question 3. Was any dispersion correction applied?

Reply: According to your kind suggestion, we have employed B3P86D3 with dispersion correction relative to bare B3P86 in calculation, yielding a calculated Co^{III} reduction potential of -1.26 V vs. NHE, which shows less consistency than that from bare B3P86 (-1.04 V vs. 0.94 V vs. NHE). These results suggest that the consideration of dispersion correction may be not suitable in this case.

Question 4. Is there any evidence that the small basis set (def2SVP) is good enough for geometry optimisation? Note that the triple zeta basis set (def2TZVP) was only used for a single-point energy correction.

Reply: According to your suggestion, we carefully thought about the suitability of the small basis set (def2SVP). It can be concluded that the small basis set (def2SVP) is sufficient and efficient to obtain reasonable results approaching to the experimental values of Co^{III} reduction (-1.04 V vs. -0.94 V). Meanwhile, a similar result (-1.02 V) can be produced with the full use of a larger basis set (def2TZVP) in parallel calculation, however consuming a much longer calculation time. Consequently, the def2SVP was employed in geometrical calculation with the use of def2TZVP in single-point calculations for optimal calculation efficiency. The discussion has been included in the section of Methods (Page 19). The details are supplied in reply to question 1.

Reviewer #3 (Remarks to the Author):

The article is worth of publication after the following issues have been addressed:

Question 1. The sections have to be rechecked. The conclusions are written in a section titled "Discussion" whereas the section titled "Results" is including also the discussion.

Reply: According to your kind suggestion, the description was edited in the main text, and the section titles were revised to “Results and discussion” and “Conclusion”, respectively. Thanks a lot.

Question 2. The analytical methods and experimental details of the reactivity assessment are not enough. The GC, the detector and the column have not been specified.

Reply: Thank you for your kind suggestion. Accordingly, the analytical methods and experimental details, such as detailed information of the gas chromatography, were supplied in the Method section in the revised manuscript (Page 15).

The details are as following:

Instruments

PXRD data were collected by a Smart X-ray diffractometer (SmartLab 9 KW, Rigaku, Japan) with Cu K α radiation ($\lambda = 1.54178 \text{ \AA}$). EDS mapping images were acquired on an Environment Scanning Electron Microscope with a field emission gun (Quanta FEG 250, FEI, America). The morphologies were recorded by a scanning electron microscope (Verios 460L, FEI, America) and a transmission electron microscope (Talos F200 X, FEI, America). The Co/Cd contents were determined by ICP-MS (iCAP RQ, Germany). Raman spectra were recorded on a high-resolution laser confocal fiber Raman spectrometer (HORIBA EVOLVTION, HORIBA Jobinyvon, France). More elemental information, especially for their valence states, was determined by XPS (ESCALAB250Xi, THERMO SCIENTIFIC, United Kingdom).

The evolved CO and H₂ were monitored by using an Agilent 7820A gas chromatography with a thermal conductivity detector (TCD) and a TDX-01 packed column, the oven temperature was held constant at 60 °C, the inlet and detector temperature were set at 80 °C and 200 °C, respectively. The gas analysis was also

operated on another gas chromatography (GC-2014+ATF, 230C, Shimadzu, Japan) equipped with two automated gas sampling valves, which contain a TCD and a flame ionization detector (FID). The carrier gas used in the above two gas chromatography equipment was argon. The fluorescent spectra were conducted on a fluorescence spectrophotometer (F-7000, Hitachi, Japan). Time-resolved fluorescence measurements were measured with MicroTime 200 time-resolved confocal fluorescence instrument. The experimental data was analyzed by the SymPhoTime 64. Electrochemical measurements were operated on a CHI660D work station.

Question 3. The assessment of the proposed catalytic system with the ones reported in literature has to be properly completed.

Reply: Thank you for the valuable suggestion. Accordingly, we have supplemented the assessment of the proposed catalytic system with more reported pioneering examples in the revised manuscript (Page 9).

Thanks a lot!

REVIEWERS' COMMENTS

Reviewer #1 (Remarks to the Author):

The authors have taken great efforts in conducting more supplementary experiments to make the conclusions more convincing. I therefore recommend this manuscript to be accepted.

Reviewer #2 (Remarks to the Author):

I have reviewed the manuscript again, and I am satisfied with the answers provided and the modifications made.

Response to Editor and Reviewers' Comments

Reviewer #1 (Remarks to the Author):

The authors have taken great efforts in conducting more supplementary experiments to make the conclusions more convincing. I therefore recommend this manuscript to be accepted.

Reviewer #2 (Remarks to the Author):

I have reviewed the manuscript again, and I am satisfied with the answers provided and the modifications made.

Reply: Thanks a lot for your nice comments that support us publishing this work in the journal of Nat. Commun.